# Plasma Omega-3 Fatty Acids and Risk for Incident Dementia in the UK Biobank Study: A Closer Look

**DOI:** 10.3390/nu15234896

**Published:** 2023-11-23

**Authors:** Aleix Sala-Vila, Nathan Tintle, Jason Westra, William S. Harris

**Affiliations:** 1The Fatty Acid Research Institute, Sioux Falls, SD 57106, USA; nlt@faresinst.com (N.T.); jason@faresinst.com (J.W.); wsh@faresinst.com (W.S.H.); 2Hospital del Mar Research Institute, 08003 Barcelona, Spain; 3Centro de Investigación Biomédica en Red de Fisiopatología de la Obesidad y Nutrición (CIBEROBN), Instituto de Salud Carlos III, 28029 Madrid, Spain; 4Department of Population Health Nursing Science, College of Nursing, University of Illinois—Chicago, Chicago, IL 60612, USA; 5Department of Internal Medicine, Sanford School of Medicine, University of South Dakota, Sioux Falls, SD 57105, USA

**Keywords:** Alzheimer’s disease, dementia, diet, fatty acids, docosahexaenoic acid, DHA, biomarkers, lipidomics

## Abstract

Dietary omega-3 fatty acids are promising nutrients in dementia. Several prospective cohort studies have examined the relationships between circulating omega-3 (an objective biomarker of dietary intake) and incident dementia, the largest to date being a report from the UK Biobank (n = 102,722). Given the recent release of new metabolomics data from baseline samples from the UK Biobank, we re-examined the association in a much larger sample (n = 267,312) and also focused on associations with total omega-3, docosahexaenoic acid (DHA), and non-DHA omega-3. Using Cox regression models, we observed that the total omega-3 status was inversely related to the risk of Alzheimer’s (Q5 vs. Q1, hazard ratio [95% confidence interval] = 0.87 [0.76; 1.00]) and all-cause dementia (Q5 vs. Q1, 0.79 [0.72; 0.87]). The strongest associations were observed for total omega-3 (and non-DHA omega-3) and all-cause dementia. In prespecified strata, we found stronger associations in men, and in those aged ≥60 years at baseline (vs. those aged 50–59). Thus, in the largest study to date on this topic, we confirmed the favorable relationships between DHA and risk for dementia, and we also found evidence that non-DHA omega-3 may be beneficial. Finally, we have better defined the populations most likely to benefit from omega-3-based interventions.

## 1. Introduction

There is an increasing body of evidence linking specific nutrients, foods and dietary patterns with Alzheimer’s disease (AD) and other dementias [1]. Most research has focused on the omega-3 fatty acid docosahexaenoic acid (DHA), which is selectively enriched and avidly retained in membrane phospholipids of the central nervous system from the third trimester of gestation [2]. There is increasing evidence that other omega-3 fatty acids, which marginally contribute to DHA levels via endogenous synthesis, may also have cognitive benefits on their own [2,3]. This is the basis for the hypothesis that an increased intake of omega-3 fatty acids might lower the risk of developing AD and other dementias, a view supported by several prospective observational studies [4].

Despite this, no primary randomized controlled trials (RCTs) have been published testing the effects of dietary omega-3 fatty acids on incident AD/dementia. Such RCTs face an array of methodological challenges (reaching the right people, with the right dose, at the right age and at the right time). In addition, the protracted evolution of dementia limits the ability of RCTs to prove causal mechanisms. Compared to RCTs, high-quality investigations using deeply phenotyped prospective cohort studies allow for the investigation of these relationships over a longer window of exposure. According to the Nutrition for Dementia Prevention Working Group, “applying biomarker tools and measures to observational studies can inform the design of new trials and encourage precision medicine” [5]. Some prospective studies have examined the association between blood omega-3 fatty acids (a biomarker of dietary intake and metabolism [6]) and incident AD and/or dementia in different prospective cohort studies worldwide [7,8,9,10,11,12,13,14]. The largest one to date is from the UK Biobank, a deeply phenotyped cohort of approximately 500,000 adults [15], in which nuclear magnetic resonance (NMR)-assessed metabolic biomarkers were made available for 117,994 participants at baseline assessment [16]. In a study with available data on omega-3 and incident dementia in 102,722 participants, both plasma DHA and total omega-3 (expressed as proportion of total fatty acids) related to a significantly lower risk of incident dementia in a model including age, sex, education, and *APOLIPOPROTEIN-E* (*APOE*)-ɛ4 status as covariates, although the statistical significance was blunted after the inclusion of sociodemographic, cardiovascular and lifestyle variables in the models [13]. Given that NMR biomarker data for approximately 300,000 individuals were released to the UK Biobank resource in July 2023, here, we re-examined the association and also explored relationships with both DHA and the non-DHA omega-3 species and how associations might differ in men and women and by the decade of life at enrollment.

## 2. Materials and Methods

### 2.1. Study Design and Participants

The UK Biobank is a prospective, population-based cohort of approximately 500,000 individuals aged between 40 and 69 years at recruitment (between 2006 and 2010). Baseline data were collected at twenty-two centers across England, Wales and Scotland [15]. Baseline data derived from questionnaires, biological samples and physical measurements were collected on all participating individuals, with longitudinal monitoring occurring via a mix of in-person and electronic medical record data [17]. The participants completed a touchscreen questionnaire, which collected information on socio-demographic characteristics, diet and lifestyle factors. Anthropometric measurements were taken using standardized procedures. The touchscreen questionnaire and other resources are shown on the UK Biobank website (http://www.ukbiobank.ac.uk; accessed on 30 October 2023). The UK Biobank has ethical approval (Ref. 11/NW/0382) from the North West Multi-centre Research Ethics Committee as a Research Tissue Bank. This approval means that researchers do not require separate ethical clearance and can operate under the Research Tissue Bank approval. All participants gave electronic signed informed consent. The UK Biobank study was conducted according to the guidelines outlined in the Declaration of Helsinki.

In July 2023, blood fatty acid data from 272,685 individuals from the UK Biobank cohort became available. We first dropped 5256 individuals with missing covariates (listed in the Statistical Methods section), leaving a sample of 267,429 individuals for this updated analysis. After dropping those with prevalent dementia at baseline (n = 117), our final analysis dataset consisted of 267,312 individuals.

### 2.2. Ascertainment of Exposure

While the phase 1 release of the UK Biobank shared metabolic biomarker data from approximately 118,000 participants at baseline recruitment (measured between June 2019 and April 2020), the phase 2 release covered metabolic biomarker data from an additional 157,000 participants at baseline recruitment (measured between April 2020 and June 2022). The phase 1 and 2 samples were both a random subset of the full cohort. A total of 249 metabolic measures were quantified in Nightingale Health’s metabolic biomarker platform based on high-throughput NMR spectroscopy [16]. The fatty acid biomarkers include DHA, omega-6 linoleic acid, total omega-3 polyunsaturated fatty acids (PUFA), total omega-6 PUFA, total PUFA, total monounsaturated fatty acids (MUFA), total saturated fatty acids (SFA), and total fatty acids. For each exposure, the metric represents a combination of fatty acids in lipid fractions (i.e., triglycerides, phospholipids, or cholesterol esters) and free fatty acids (also called non-esterified fatty acids) [6]. Both the concentration of each fatty acid and the corresponding percentage (by weight) of total fatty acids were calculated. In our study, the exposures of interest were the “Docosahexaenoic Acid to Total Fatty Acids percentage” (DHA%) and “Omega-3 Fatty Acids to Total Fatty Acids percentage” (n3%). Given that individual omega-3 species other than DHA are not available in the UK Biobank, to examine whether omega-3 species other than DHA are associated with incident dementia and AD, we calculated “Non-DHA omega-3 fatty acids” by subtracting the DHA value from the total omega-3 value (non-DHA n3%).

### 2.3. Ascertainment of Outcomes

Incident dementia cases were ascertained using data linkage to hospital inpatient records (Hospital Episode Statistics for England, Morbidity Records for Scotland, and the Patient Episode Database for Wales) and death register data (National Health Service (NHS) Digital, NHS Central Register, and National Records). Incident AD was defined by code 331.0 in the International Classification of Diseases 9th revision (ICD-9) and codes F00 (including atypical or mixed type) and G30 in ICD-10. All-cause dementia was defined as all of the prior codes plus ICD-9 codes 290, 291.2, 294.1, 331.0–331.2, and 331.5 and ICD-10 codes I67.3, F01, A81.0, F02, F05.1, F10.6, G31.0, G31.1, and G31.8. The censoring time of incident dementia in our study was 11 December 2021. For each participant, the time to event was calculated as months from the date of baseline to the date of the first diagnosis of dementia, date of death, date of loss to follow-up, or censoring time, whichever came first.

### 2.4. Statistical Analyses

Sample characteristics were summarized using standard metrics. As in prior studies, we focused primarily on modeling the relationships between exposures of interest and the two outcomes with quintiles (with the lowest quintile as the reference) and continuous linear relationships (per IQ_5_R analysis, defined as 90th minus 10th percentiles of each exposure of interest). We constructed Cox proportional hazards adjusting for gender, age at enrollment, *APOE*-ε4 carriership (non-carrier vs. carrier), education (college or above vs. high school degree or equivalent vs. less than high school vs. unknown), body mass index (BMI), Townsend deprivation index, and prevalent diabetes at baseline (yes vs. no).

In order to assess the interaction of each exposure of interest with the decade of life at enrollment (including gender, *APOE*-ε4 carriership, education, BMI, Townsend deprivation index, and prevalent diabetes at baseline as covariates), we added the corresponding interaction term with fatty acid to each model. Alternatively, we examined interactions of the exposure of interest with *APOE*-ε4 carriership (including gender, age at enrollment, education, BMI, Townsend deprivation index, and prevalent diabetes at baseline as covariates). Finally, we assessed associations in prespecified strata by gender and by decade of life at enrollment. We anticipated that a low number of participants aged 40–49 at enrollment would develop dementia during the follow-up, so we did not include them in the age interaction analysis and stratified analysis. All analyses were conducted in R version 4.1.2 and used a statistical significance threshold of 0.05.

## 3. Results

The mean (SD) baseline proportions of total n3%, DHA%, and non-DHA n3% were 4.38 (1.55), 2.00 (0.68), and 2.39 (1.01), respectively. Racially, the study population was 95% White. Covariates included in the models are presented in Table 1.

During the follow-up, 2280 cases of AD (1091 in men; 1189 in women) and 5193 cases of all-cause dementia (2740 in men; 2453 in women) were ascertained. A total of 17 cases of AD and 72 cases of all-cause dementia were reported in the group of participants aged 40–49 at baseline. No statistically significant interactions between any exposure of interest and *APOE*-ε4 carriership were observed. Table 2 displays HRs for plasma total omega-3 fatty acids and incident AD, both in the whole study population and after stratifying by gender and age at enrollment.

Compared to participants at the lowest quintile, those at the top quintile had a 13% reduction in risk for incident AD (*p* = 0.04), with a much stronger association in men (*p* = 0.009) than in women (*p* = 0.727). Regarding age strata (*p* interaction = 0.401), we observed significant associations in those aged ≥ 60 years at baseline. For continuous analysis (per IQ_5_R), statistically significant reductions were restricted to men (*p* = 0.0083). The test for interaction per IQ_5_R provided evidence for an interaction effect with age (*p* = 0.023).

We observed much stronger associations for incident all-cause dementia than AD (Table 3). Compared to participants in Q1, those in the upper quintiles had a significantly lower risk when considering the full cohort, only men, and only those aged ≥ 60 years at baseline. The *p* value for interaction with age was 0.366. Associations were similar in direction and magnitude in per IQ_5_R analyses, yet with a statistically significant interaction with age (*p* = 0.038), with a stronger association in those aged 50–59 at baseline. In women, a significantly lower risk was observed for those in Q3, Q4, or Q5 compared to Q1.

We next examined the associations for DHA%. When considering incident AD as the outcome of interest (Table 4), significantly lower risks were limited to participants aged 50–59 years at baseline (Q3, Q4 and Q5 vs. Q1; per IQ_5_R analysis). Statistically significant interactions were observed with age, both in the per quintile and per IQ_5_R analyses (*p* = 0.004 and *p* < 0.001, respectively).

Once again, we observed stronger associations with incident all-cause dementia (Table 5). Compared to participants in Q1, those in the upper quintiles had a significantly lower risk when considering the full cohort and only women. Likewise, a statistically significant lower risk was observed for men (Q5 vs. Q1) and for participants aged ≥ 60 years at baseline (Q3, Q4 and Q5 vs. Q1) (*p* interaction with age = 0.003). When exposure was examined as a continuous variable, significant inverse associations were observed when considering the entire cohort, men only, and in both age strata (*p* interaction with age = 0.006).

Finally, we considered non-DHA n3% as an exposure. For incident AD (Table 6), significantly lower risk was observed in the whole cohort, in men, and in those aged ≥ 60 years at baseline, both in per quintile analysis (significant reductions in Q4 and Q5 vs. Q1) and in per IQ_5_R. We found no evidence of interaction with age. We observed lower risks for incident dementia (Table 7) in the same groups (Q2, Q3, Q4, Q5 vs. Q1, and per IQ_5_R). Once again, we found no evidence of interaction with age.

## 4. Discussion

In this study, we updated the associations between blood omega-3 biomarkers and incident dementia in the framework of the UK Biobank [13] after the recent release of metabolic biomarker data from an additional 157,000 participants at baseline recruitment (n = 267,312; the largest prospective study on the topic to date). We considered three different exposures (n3%, DHA% and non-DHA n3%) and two different outcomes (incident AD and all-cause dementia), and we stratified for gender and age at baseline (50 to 59 years vs. ≥60 years). Most of the associations with incident disease were inverse regardless of the exposure or the outcome. However, the strongest associations were observed for n3% and non-DHA n3% and incident all-cause dementia. In analyses after stratification, in general terms, associations were stronger in men than in women, and stronger in those ≥ 60 years at baseline than in younger participants.

This work was intended to provide information to help guide the design of future long-term interventions with omega-3 fatty acids for the prevention of AD and other dementias. The design of such RCTs would naturally require the consideration of many methodological issues. One of the most important (and controversial) issues is which omega-3 fatty acid(s) to test. While the strength of the association of DHA intake with dementia from experimental and epidemiologic studies appears to be clinically relevant, there is the long-standing question of whether other omega-3 fatty acids (such as eicosapentaenoic acid (EPA), docosapentaenoic acid (DPA), and alpha-linolenic acid (ALA)) may also play preventive roles in dementia. Given that only data on DHA and total omega-3 fatty acids are available for the UK Biobank, we created a novel variable of “non-DHA n3%” as the best proxy for the other fatty acids in this family since further granularity was not possible. Interestingly, we found that associations for non-DHA n3% were generally stronger than those observed for DHA% when considering both AD and all-cause dementia as the outcomes of interest. This finding is aligned with the results of other prospective studies analyzing different omega-3 biomarker species, which reported lower risks for EPA [8,11] and DPA [14] than for DHA. This reinforces the notion that, although DHA is the main omega-3 in brain tissues, other dietary omega-3 might also play a role in the development of dementias, either through conversion to DHA or, more plausibly, by providing benefits on their own, as increasingly seen in experimental research studies [3,18,19].

We also examined sex differences, which are increasingly recognized as a key priority in research and clinical development in this field [20,21]. Although women represent two-thirds of individuals with AD [22], they have long been under-represented in many RCTs and, when included, the typically small numbers reduced statistical power, resulting in imprecise effect estimates, possibly missing potential benefits. In sex-stratified analyses, we observed that inverse associations between omega-3 and incident dementia were weaker for women than for men, and statistically significant lower risks were mostly restricted to all-cause dementia rather than to AD. Such differences between genders regarding the magnitude of the association warrant further investigation, particularly if we consider the current lack of cognitive evidence for sex differences from dietary intervention studies [23].

Finally, we searched for associations after stratifying for age (decade of life) at baseline. Dementia progresses slowly, with detectable brain alterations starting up to 20 years before the onset of clinical symptomatology [24]. This prodromal phase would be the most logical period in which to institute therapies to slow the progression of cognitive loss. Although there is an interest in how lifestyle in middle age affects later health, almost all prospective studies on omega-3 and incident dementias have been conducted in populations aged ≥65 years old [7,8,9,10,11,12,14]. This is because of the low prevalence of dementia before 65 years of age (for example, in Europe, the prevalence is 0.6% in the age range between 60 and 64 years [22]). The number of individuals needed for such trials would be very large, probably rendering such trials unfundable. Such a concern was, to some extent, circumvented by the large sample size of our study (n = 88,881 participants aged 50 to 59 years at baseline, with 210 and 607 ascertained cases of AD and all-cause dementia, respectively). We observed lower risks in this specific population, although statistically significant associations were restricted to n3% and all-cause dementia, to DHA% and AD, and to DHA% and all-cause dementia. Stronger associations were observed in those who were in their 60s at enrollment. Further research should explore whether such differences in magnitude (but not in direction) of the associations between the two age groups are explained by either a real underlying difference in mechanisms or by differences in statistical power since the group of participants aged ≥ 60 years at baseline was significantly larger and had a much greater incidence of AD and all-cause dementia.

## 5. Conclusions

In conclusion, in the largest prospective cohort study to date on omega-3 biomarkers and incident dementia, we observed that increasing proportions of these fatty acids in blood are inversely related to the risk of suffering this devastating disease. Stronger associations were observed for non-DHA n3% compared to DHA%; for all-cause dementia compared to AD (suggestive that the effect of omega-3 is likely greater for non-AD dementias, such as vascular dementia, Lewy body dementia, and frontotemporal dementia); for men compared to women; and for those over age 60 at baseline compared to those in their 50s. This research provides evidence for the benefits of omega-3 fatty acids in brain health and contributes to better defining populations who might obtain the greatest cognitive benefits in omega-3-based interventions.

## Figures and Tables

**Table 1 nutrients-15-04896-t001:** Characteristics of the study population (n = 267,312).

Variable ^1^	
Women—No. (%)	144,487 (54)
Age at baseline—y	56.5 (8.1)
40–49 years—No. (%)	62,324 (23.3)
50–59 years—No. (%)	88,881 (33.2)
≥60 years—No. (%)	116,107 (43.4)
Education—No. (%)	
College or above	129,434 (48.4)
High school or equivalent	90,676 (33.9)
Less than high school	47,202 (17.7)
Body mass index—kg/m^2^	27.4 (4.8)
Diabetes—No. (%)	14,047 (5.3)
Townsend deprivation index	−1.38 (3.06)

^1^ Data are expressed as mean (95% SD), except for categorical variables (expressed as N and %).

**Table 2 nutrients-15-04896-t002:** Hazard ratios (HR) for plasma total omega-3 fatty acids and incident Alzheimer’s disease (n = 267,312).

Stratification		Per Quintiles					Per IQ_5_R
		Q1 (<3.15, Median = 2.68)	Q2 (3.15 to 3.85, Median = 3.52)	Q3 (3.85 to 4.52, Median = 4.17)	Q4 (4.52 to 5.45, Median = 4.92)	Q5 (>5.45, Median = 6.32)	
All	Cases/n	366/53,462	416/53,456	462/53,467	482/53,463	554/53,464	2280/267,312
	HR (95% CI)	1.00	0.99 (0.86; 1.14)	0.94 (0.82; 1.08)	0.88 (0.76; 1.01)	0.87 (0.76; 1.00) *	0.92 (0.83; 1.01)
Gender	**Men**						
	Cases/n	229/29,204	233/27,128	236/25,201	203/22,170	190/19,122	1091/122,825
	HR (95% CI)	1.00	0.97 (0.80; 1.17)	0.91 (0.76; 1.10)	0.83 (0.69; 1.01)	0.77 (0.63; 0.94) **	0.82 (0.71; 0.95) **
	**Women**						
	Cases/n	137/24,258	183/26,328	226/28,266	279/31,293	364/34,342	1189/144,487
	HR (95% CI)	1.00	1.03 (0.82; 1.29)	0.99 (0.80; 1.24)	0.94 (0.76; 1.16)	0.97 (0.79; 1.18)	0.99 (0.87; 1.13)
Age	**50–59 years**						
	Cases/n	50/18,495	50/18,572	46/17,956	32/17,495	32/13,363	210/88,881
	HR (95% CI)	1.00	1.02 (0.68; 1.53)	1.01 (0.67; 1.52)	0.74 (0.47; 1.17)	0.77 (0.48; 1.22)	0.71 (0.49; 1.03)
	**≥60 years**						
	Cases/n	313/17,316	359/19,725	413/22,987	449/26,056	519/30,023	2053/116,107
	HR (95% CI)	1.00	0.98 (0.83; 1.14)	0.93 (0.80; 1.08)	0.89 (0.76; 1.03)	0.87 (0.75; 1.00) *	0.93 (0.83; 1.03)

Note: CI = confidence interval; IQ_5_R, interquintile range (defined as 90th minus 10th percentiles of circulating total omega-3); Q = quintile. Adjusted for gender (except in gender stratification), age at enrollment (except in age stratification), *APOE*-ε4 carriership (non-carrier vs. carrier), education (college or above vs. high school degree or equivalent vs. less than high school vs. unknown), BMI, Townsend deprivation index, and prevalent diabetes at baseline (yes vs. no). *, *p* < 0.05; **, *p* < 0.01.

**Table 3 nutrients-15-04896-t003:** Hazard ratios (HR) for plasma total omega-3 fatty acids and incident all-cause dementia (n = 267,312).

Stratification		Per Quintiles					Per IQ_5_R
		Q1 (<3.15, Median = 2.68)	Q2 (3.15 to 3.85, Median = 3.52)	Q3 (3.85 to 4.52, Median = 4.17)	Q4 (4.52 to 5.45, Median = 4.92)	Q5 (>5.45, Median = 6.32)	
All	Cases/n	938/53,462	938/53,456	1053/53,647	1073/53,463	1191/53,464	5193/267,312
	HR (95% CI)	1.00	0.88 (0.81; 0.97) **	0.87 (0.79; 0.95) **	0.82 (0.74; 0.89) ***	0.79 (0.72; 0.87) ***	0.87 (0.81; 0.93) ***
Gender	**Men**						
	Cases/n	6120/29,204	564/27,128	585/25,201	509/22,170	470/19,122	2740/122,825
	HR (95% CI)	1.00	0.89 (0.79; 1.00) *	0.88 (0.78; 0.98) *	0.81 (0.72; 0.92) ***	0.74 (0.66¸0.84) ***	0.80 (0.73; 0.88) ***
	**Women**						
	Cases/n	326/24,258	374/26,328	468/28,266	564/31,293	721/34,342	2453/144,487
	HR (95% CI)	1.00	0.88 (0.76; 1.03)	0.87 (0.75; 1.00) *	0.83 (0.72; 0.95) **	0.83 (0.73; 0.96) **	0.93 (0.84; 1.03)
Age	**50–59 years**						
	Cases/n	158/18,495	121/18,572	124/17,956	98/17,495	100/13,363	601/88,881
	HR (95% CI)	1.00	0.79 (0.62; 1.00) *	0.87 (0.68;1.11)	0.73 (0.56; 0.95) *	0.81 (0.622; 1.05)	0.81 (0.65; 1.00) *
	**≥60 years**						
	Cases/n	758/17,316	798/19,725	914/22,987	967/26,056	1083/30,023	4520/116,107
	HR (95% CI)	1.00	0.90 (0.81; 1.00) *	0.87 (0.79; 0.96) **	0.73 (0.75; 0.91) ***	0.79 (0.72; 0.87) ***	0.87 (0.81; 0.93) ***

Note: CI = confidence interval; IQ_5_R, interquintile range (defined as 90th minus 10th percentiles of circulating total omega-3); Q = quintile. Adjusted for gender (except in gender stratification), age at enrollment (except in age stratification), *APOE*-ε4 carriership (non-carrier vs. carrier), education (college or above vs. high school degree or equivalent vs. less than high school vs. unknown), BMI, Townsend deprivation index, and prevalent diabetes at baseline (yes vs. no). *, *p* < 0.05; **, *p* < 0.01; ***, *p* < 0.001.

**Table 4 nutrients-15-04896-t004:** Hazard ratios (HR) for plasma DHA and incident Alzheimer’s disease (n = 267,312).

Stratification		Per Quintiles					Per IQ_5_R
		Q1 (<1.46, Median = 1.24)	Q2 (1.46 to 1.78, Median = 1.63)	Q3 (1.78 to 2.07, Median = 1.92)	Q4 (2.07 to 2.47, Median = 2.25)	Q5 (>2.47, Median = 2.84)	
All	Cases/n	429/53,457	404/53,455	431/53,459	484/53,473	532/53,468	2280/267,312
	HR (95% CI)	1.00	0.92 (0.80; 1.06)	0.91 (0.79; 1.04)	0.94 (0.82; 1.07)	0.93 (0.81; 1.06)	0.98 (0.89; 1.09)
Gender	**Men**						
	Cases/n	267/33,633	220/27,288	205/23,521	204/20,240	195/18,143	1091/122,825
	HR (95% CI)	1.00	0.96 (0.80; 1.15)	0.93 (0.78; 1.13)	0.95 (0.79; 1.15)	0.91 (0.75: 1.10)	0.92 (0.80; 1.06)
	**Women**						
	Cases/n	162/19,824	184/26,167	226/29,938	280/33,233	337/35,325	1189/144,487
	HR (95% CI)	1.00	0.88 (0.71; 1.10)	0.88 (0.72; 1.09)	0.92 (0.76; 1.13)	0.94 (0.77: 1.14)	1.06 (0.92; 1.12)
Age	**50–59 years**						
	Cases/n	70/18,871	43/18,288	33/17,880	34/17,213	30/16,629	210/88,881
	HR (95% CI)	1.00	0.71 (0.48; 1.05)	0.58 (0.38; 0.89) *	0.65 (0.42; 1.00) *	0.59 (0.37; 0.94) *	0.62 (0.42; 0.90) *
	**≥60 years**						
	Cases/n	351/20,135	360/24,461	397/22,540	445/24,457	500/27,514	2053/116,107
	HR (95% CI)	1.00	0.98 (0.85; 1.14)	0.99 (0.85; 1.14)	0.99 (0.86; 1.15)	0.99 (0.85; 1.15)	1.02 (0.92; 1.13)

Note: CI = confidence interval; IQ_5_R, interquintile range (defined as 90th minus 10th percentiles of circulating DHA); Q = quintile. Adjusted for gender (except in gender stratification), age at enrollment (except in age stratification), *APOE*-ε4 carriership (non-carrier vs. carrier), education (college or above vs. high school degree or equivalent vs. less than high school vs. unknown), BMI, Townsend deprivation index, and prevalent diabetes at baseline (yes vs. no). *, *p* < 0.05.

**Table 5 nutrients-15-04896-t005:** Hazard ratios (HR) for plasma DHA and incident all-cause dementia (n = 267,312).

Stratification		Per Quintiles					Per IQ_5_R
		Q1 (<1.46, Median = 1.24)	Q2 (1.46 to 1.78, Median = 1.63)	Q3 (1.78 to 2.07, Median = 1.92)	Q4 (2.07 to 2.47, Median = 2.25)	Q5 (>2.47, Median = 2.84)	
All	Cases/n	1065/53,457	973/53,455	992/53,459	1039/53,473	1124/53,468	5193/267,312
	HR (95% CI)	1.00	0.91 (0.83; 1.00) *	0.88 (0.81; 0.97) **	0.88 (0.80; 0.86) **	0.87 (0.79; 0.95) **	0.91 (0.86; 0.98) **
Gender	**Men**						
	Cases/n	695/33,633	583/27,288	515/23,521	480/20,240	467/18,143	2740/122,825
	HR (95% CI)	1.00	0.98 (0.88; 1.10)	0.93 (0.82; 1.04)	0.90 (0.80; 1.01)	0.87 (0.77; 0.99) *	0.88 (0.80; 0.96) **
	**Women**						
	Cases/n	370/19,824	390/26,167	477/29,938	559/33,233	657/35,325	2453/144,487
	HR (95% CI)	1.00	0.82 (0.71; 0.94) **	0.82 (0.72; 0.95) **	0.84 (0.73; 0.96) *	0.84 (0.73: 0.96) *	0.97 (0.88; 1.07)
Age	**50–59 years**						
	Cases/n	175/18,871	130/18,288	104/17,880	98/17,213	94/16,629	601/88,881
	HR (95% CI)	1.00	0.87 (0.69; 1.10)	0.75 (0.58; 0.96) *	0.79 (0.61; 1.03)	0.80 (0.61; 1.05)	0.81 (0.65; 1.00) *
	**≥60 years**						
	Cases/n	862/20,135	834/21,461	869/22,540	932/24,457	1023/27,514	4520/116,107
	HR (95% CI)	1.00	0.94 (0.85; 1.04)	0.91 (0.83; 1.00) *	0.90 (0.82; 1.00) *	0.89 (0.81; 0.98) *	0.93 (0.87; 1.00) *

Note: CI = confidence interval; IQ_5_R, interquintile range (defined as 90th minus 10th percentiles of circulating DHA); Q = quintile. Adjusted for gender (except in gender stratification), age at enrollment (except in age stratification), *APOE*-ε4 carriership (non-carrier vs. carrier), education (college or above vs. high school degree or equivalent vs. less than high school vs. unknown), BMI, Townsend deprivation index, and prevalent diabetes at baseline (yes vs. no). *, *p* < 0.05; **, *p* < 0.01.

**Table 6 nutrients-15-04896-t006:** Hazard ratios (HR) for plasma non-DHA omega-3 fatty acids and incident Alzheimer’s disease (n = 267,312).

Stratification		Per Quintiles					Per IQ_5_R
		Q1 (<1.57, Median = 1.20)	Q2 (1.57 to 2.08, Median = 1.84)	Q3 (2.08 to 2.53, Median = 2.30)	Q4 (2.53 to 3.13, Median = 2.80)	Q5 (>3.13, Median = 3.65)	
All	Cases/n	354/53,462	398/53,463	499/53,455	482/53,468	547/53,464	2280/267,312
	HR (95% CI)	1.00	0.92 (0.79; 1.06)	0.99 (0.86; 1.14)	0.84 (0.73; 0.97) *	0.82 (0.72; 0.95) **	0.88 (0.79; 0.97) *
Gender	**Men**						
	Cases/n	214/26,087	222/26,133	260/25,751	197/23,986	198/20,868	1091/122,825
	HR (95% CI)	1.00	0.93 (0.77; 1.12)	0.98 (0.81; 1.18)	0.74 (0.61; 0.90) **	0.75 (0.61; 0.91) **	0.77 (0.66; 0.90) ***
	**Women**						
	Cases/n	140/27,375	176/27,330	239/27,704	285/29,482	349/32,596	1189/144,487
	HR (95% CI)	1.00	0.89 (0.71; 1.12)	1.02 (0.82; 1.26)	0.93 (0.75; 1.14)	0.89 (0.72; 1.08)	0.96 (0.83: 1.10)
Age	**50–59 years**						
	Cases/n	43/18,191	42/18,505	47/18,220	46/17,664	32/16,301	210/88,881
	HR (95% CI)	1.00	0.93 (0.60; 1.44)	1.08 (0.71; 1.66)	1.08 (0.70; 1.66)	0.79 (0.49; 1.26)	0.81 (0.57; 1.17)
	**≥60 years**						
	Cases/n	306/16,190	353/19,995	447/23137	435/26,378	512/30,407	2053/116,107
	HR (95% CI)	1.00	0.91 (0.78; 1.07)	0.98 (0.84; 1.13)	0.82 (0.70; 0.95) **	0.81 (0.70; 0.94) **	0.87 (0.78; 0.97) *

Note: CI = confidence interval; IQ_5_R, interquintile range (defined as 90th minus 10th percentiles of circulating non-DHA omega-3); Q = quintile. Adjusted for gender (except in gender stratification), age at enrollment (except in age stratification), *APOE*-ε4 carriership (non-carrier vs. carrier), education (college or above vs. high school degree or equivalent vs. less than high school vs. unknown), BMI, Townsend deprivation index, and prevalent diabetes at baseline (yes vs. no). *, *p* < 0.05; **, *p* < 0.01; ***, *p* < 0.001.

**Table 7 nutrients-15-04896-t007:** Hazard ratios (HR) for plasma non-DHA omega-3 fatty acids and incident all-cause dementia (n = 267,312).

Stratification		Per Quintiles					Per IQ_5_R
		Q1 (<1.57, Median = 1.20)	Q2 (1.57 to 2.08, Median = 1.84)	Q3 (2.08 to 2.53, Median = 2.30)	Q4 (2.53 to 3.13, Median = 2.80)	Q5 (>3.13, Median = 3.65)	
All	Cases/n	354/53,462	398/53,463	499/53,455	482/53,468	547/53,464	5193/267,312
	HR (95% CI)	1.00	0.86 (0.78; 0.94) **	0.89 (0.81; 0.98) *	0.80 (0.73; 0.88) ***	0.79 (0.73; 0.87) ***	0.85 (0.79; 0.91) ***
Gender	**Men**						
	Cases/n	574/26,087	537/26,133	598/25,751	517/23,986	514/50,868	2740/122,825
	HR (95% CI)	1.00	0.85 (0.76; 0.96) **	0.86 (0.76; 0.96) **	0.74 (0.66; 0.84) ***	0.75 (0.66; 0.85) ***	0.77 (0.70; 0.85) ***
	**Women**						
	Cases/n	311/27,375	371/27,330	487/27,704	574/29,482	710/32,596	2453/144,487
	HR (95% CI)	1.00	0.86 (0.74; 1.00) *	0.95 (0.82; 1.10)	0.88 (0.76: 1.04)	0.85 (0.74; 0.97) *	0.91 (0.83; 1.01)
Age	**50–59 years**						
	Cases/n	146/18,191	116/18,505	116/18,220	116/17,664	107/16,301	601/88,881
	HR (95% CI)	1.00	0.77 (0.60; 0.99) *	0.80 (0.62; 1.02)	0.81 (0.63; 1.05)	0.81 (0.53; 1.05)	0.83 (0.67; 1.03)
	**≥60 years**						
	Cases/n	715/16,190	775/19,995	956/23,137	967/26,378	1107/23,407	4520/116,107
	HR (95% CI)	1.00	0.87 (0.79; 0.97) **	0.91 (0.82; 1.00) *	0.81 (0.73; 0.89) ***	0.79 (0.72; 0.87) ***	0.84 (0.78; 0.91) ***

Note: CI = confidence interval; IQ_5_R, interquintile range (defined as 90th minus 10th percentiles of circulating non-DHA omega-3); Q = quintile. Adjusted for gender (except in gender stratification), age at enrollment (except in age stratification), *APOE*-ε4 carriership (non-carrier vs. carrier), education (college or above vs. high school degree or equivalent vs. less than high school vs. unknown), BMI, Townsend deprivation index, and prevalent diabetes at baseline (yes vs. no). *, *p* < 0.05; **, *p* < 0.01; ***, *p* < 0.001.

## Data Availability

The codebook and analytic code related to data described in the manuscript will be made available upon reasonable request pending application to and approval by the Fatty Acid Research Institute. Raw data are available via standard application procedures directly from the UK Biobank.

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
