# Peer review of "Plasma Omega-3 Fatty Acids and Risk for Incident Dementia in the UK Biobank Study: A Closer Look"

_nutrients, 2023, doi:10.3390/nu15234896_

Round 1

Reviewer 1 Report

Comments and Suggestions for Authors

Sala-Vila et al. analyzed the relationship between plasma ω3 fatty acids such as DHA and dementia such as Alzheimer's disease using data published by the UK Biobank. The results showed that the association between ω3 fatty acids and dementia may differ depending on age and gender. The present study is very interesting because it is the result of a large-scale data analysis.

The metabolome data from the UK Biobank is highly reproducible because it was measured using NMR. However, NMR cannot identify individual molecules. For example, DHA is an aggregate of various molecules including DHA free in plasma, DHA covalently bound to glycerophospholipids and triglycerides as side chains, DHA bound to cholesterol, and so on. It is not simple to consider the uptake of these as nutrients or their transfer from the blood to the brain. The authors should comment on this point.

Since APOE is closely related to lipids and APOE4 is the strongest risk factor for Alzheimer's disease, it should be mentioned whether there are differences in the effects of ω3 fatty acids such as DHA with and without APOE4 carriers.

It should be noted that the statistical results differ between Alzheimer's disease and dementia, although Alzheimer's disease is said to account for about 60% of all dementias, and what kind of dementia has a greater impact on the statistical results should be mentioned.

Table 2-6 all have the same number of n despite the different extraction conditions. Q1-Q5 should have the same number of n, but only Q2 in Tables 6 and 7 has about 10,000 fewer than the others. The reasons for these differences should be commented on.

Author Response

We appreciate the thorough review done to our manuscript. Please see the attachment, where you can find point-by-point answers to all issues and comments you raised. Particular changes incorporated into the original manuscript here appear in green lettering to facilitate the review process.

Reviewer 2 Report

Comments and Suggestions for Authors

The manuscript does not specify which omega-3 acids, apart from DHA, were tested. Which specific acids were defined by the authors as "non-DHA omega-3"? The introduction and discussion do not provide information on the impact of individual acids on health, including dementia and Alzheimer's disease.

Moreover, the "Materials and methods" section does not precisely describe the method of determining individual fatty acids. The results of the content of individual fatty acids should be collected and presented in a supplementary materials.

The content of fatty acids in the human body (including DHA) is influenced by dietary supply, possible supplementation, as well as endogenous synthesis (enzyme activity). Meanwhile, the authors do not provide any data on this subject. This needs to be supplemented.

Not all abbreviations are explained in the text, and the “References” part was prepared contrary to the journal's guidelines.

Author Response

(The authors gave the same response as above.)

Round 2

Reviewer 1 Report

Comments and Suggestions for Authors

I have already accepted the manuscript.

Reviewer 2 Report

Comments and Suggestions for Authors

Accept in present form.